# OGG1 in Lung—More than Base Excision Repair

**DOI:** 10.3390/antiox11050933

**Published:** 2022-05-09

**Authors:** Xiaodi Ma, Hewei Ming, Lexin Liu, Jiahui Zhu, Lang Pan, Yu Chen, Yang Xiang

**Affiliations:** 1School of Basic Medicine, Central South University, Changsha 410078, China; mxd0324@csu.edu.cn (X.M.); 2111210119@bjmu.edu.cn (H.M.); llx0629@csu.edu.cn (L.L.); zhujiahui@csu.edu.cn (J.Z.); panl234@csu.edu.cn (L.P.); 2Department of Medical Laboratory, School of Medicine, Hunan Normal University, Changsha 410013, China; chenyu0213@hunnu.edu.cn; 3Department of Physiology, School of Basic Medicine, Central South University, Changsha 410008, China

**Keywords:** oxidative damage, OGG1, pulmonary diseases

## Abstract

As the organ executing gas exchange and directly facing the external environment, the lungs are challenged continuously by various stimuli, causing the disequilibration of redox homeostasis and leading to pulmonary diseases. The breakdown of oxidants/antioxidants system happens when the overproduction of free radicals results in an excess over the limitation of cleaning capability, which could lead to the oxidative modification of macromolecules including nucleic acids. The most common type of oxidative base, 8-oxoG, is considered the marker of DNA oxidative damage. The appearance of 8-oxoG could lead to base mismatch and its accumulation might end up as tumorigenesis. The base 8-oxoG was corrected by base excision repair initiated by 8-oxoguanine DNA glycosylase-1 (OGG1), which recognizes 8-oxoG from the genome and excises it from the DNA double strand, generating an AP site for further processing. Aside from its function in DNA damage repairment, it has been reported that OGG1 takes part in the regulation of gene expression, derived from its DNA binding characteristic, and showed impacts on inflammation. Researchers believe that OGG1 could be the potential therapy target for relative disease. This review intends to make an overall summary of the mechanism through which OGG1 regulates gene expression and the role of OGG1 in pulmonary diseases.

## 1. Introduction

Redox reaction exists widely in the manifold biochemical processes of the human body, rendering regular physiological function. Redox regulation could be achieved via controlling the activity of enzymes and transcriptional factors. Oxidative modification of the protein is the essential mechanism of redox signaling; moreover, sensing of the redox state is also part of redox regulation [1,2].

Redox homeostasis is generally crucial for life and is more of a dynamic balance. Free radicals, mainly reactive oxygen species (ROS) and reactive nitrogen species (RNS), produced under physiological conditions are detoxified and scavenged by antioxidants, thus maintaining at a moderate level [3]. Exposure to endogenous and exogenous stimulation, such as chemicals, radiation and microbiome, would disequilibrate the oxidant/antioxidant system, leading to the excessive production of free radicals to a level exceeding the antioxidant capacity, thus resulting in oxidative stress.

The lungs are one of the organs directly in contact with the external environment and are often stimulated by external factors. Being the first interface between the internal environment and the outside world, airway epithelium is the primary target of inhaling gas or particles, which has important physiological functions in the process of sensing an injury signal, innate immune defense and regulating inflammatory response [4]. It’s reasonable to stress the impact oxidative stress has on the development of pulmonary diseases.

Oxidative stress pervades all principal levels, causing damages ranging from oxidative DNA damage to protein oxidation. Under physiological conditions, ROS could act as signaling factors in certain circumstances, regulating gene expression and affecting biological processes, such as proliferation, migration and angiogenesis. While excessively produced ROS could attack macromolecules such as proteins, lipids and nucleic acids, the accumulation of oxidative modification might turn into disease. With its broad involvement, oxidative stress was universally acknowledged as taking part in the genesis of pathological processes, such as tumor growth [5], metastasis [6], inflammation and fibrosis [7], to which a series of diseases, such as Parkinson’s syndrome [8,9], Alzheimer’s disease [10,11], aging [12], cancer [13,14,15], obesity [16,17], cardiovascular diseases [18], pulmonary fibrosis [19], inflammatory [20], rheumatoid arthritis [21] and so on, are related (Figure 1). 

When it comes to the DNA oxidative damage, the function of 8-oxoguanine DNA Glycosylase (OGG1) is non-negligible, since 8-oxoG is the most common DNA oxidative damage and was fixed by base excision repair initiated by OGG1. This review collected and sorted evidence over the biological function of OGG1 and its new role as a regulator, trying to dig deep into the connection between OGG1 and pulmonary disease and figuring out its potential possibility to work as target therapy in the future. 

## 2. The Role of OGG1 in DNA Oxidative Modification

Oxidative stress can exacerbate airway inflammatory responses by which asthma, chronic obstructive pulmonary disease (COPD) and other invasive pulmonary diseases are accompanied [22,23]. Free radicals and reactive oxygen species can attenuate the mucosal function of organs, increase endothelial permeability, reduce endothelial adhesion and affect the reconstruction of extracellular matrix. In addition to the direct action of oxidants, the oxidative stress response can also aggravate the inflammatory response through the following mechanisms: (1). Oxidants can weaken the deformation ability of neutrophils, which causes the retention and activation of neutrophils in pulmonary microcirculation [24]; (2). Oxidative stress activates transcription factor nuclear factor κB (NF-κB) and activator protein 1 (AP-1), which regulate the release of inflammatory mediators, aggravating inflammation [25]; (3). Oxidants can promote the expression of adhesion molecules on the surface of neutrophils. Neutrophils recruited in the lungs release active oxygen and protein amines after activation, causing tissue damage at the site of inflammation [26]. Currently, most studies focus on changes in protein function or abnormal lipid metabolism under oxidative stress, whereas the mechanism of DNA oxidative modification on inflammatory response is not clear.

In the course of evolution, organisms evolved complex repair pathways that maintain genome integrity and accuracy, such as base excision repair (BER), nucleotide excision repair (NER), mismatch repair (MMR), homologous recombination (HR) and non-homologous termination (NHEJ) [27,28,29,30,31]. When facing stimulation, DNA oxidative damage could still accumulate despite the existence of multiple repair pathways, causing gene mutations or cell death [32,33]. 

DNA is oxidized to produce oxidative damage leading to modified bases (oxidized bases), chain break (DNA single and double chain break) and chain cross [34]. DNA oxidative modification has site specificity. Among all four bases—cytosine, guanine, adenine and thymine, the lowest redox potential makes guanine (G) the most fragile base for oxidative stress and 7, 8-dihydro-8-oxoguanine (8-oxoG) the most common DNA oxidative damage; thus, the appearance and accumulation of 8-oxoG in DNA sequence was also considered as the biomarker of DNA oxidative damage. As we mentioned above, 8-oxoG accumulation is usually closely related to many physiological processes, and due to the stacking of π-π bonds and the interaction of electron orbitals, the ionization energy of 5’ terminal guanine in continuous guanine is reduced, so that the 5’ ends containing multiple adjacent guanines, for example, 5’G < 5’GG < 5’GGG, could be the site of 8-oxoG preferentially [35]. The DNA double helix is not affected by 8-oxoG, but it can mismatch with adenine (A) and appear GC→TA mutation in DNA replication [36]. 

In eukaryotes, this DNA oxidative modification can be identified by the DNA glycosylation enzyme, which can repair 8-oxoG by means of base excision repair (BER) [37]. As the initiator of BER process, OGG1 plays a pivotal role in the removal of 8-oxoG and formamidopyrimidine (Fapy-G), the ring-opened guanine [37].

The base excision repair process could be divided into two sections, the AP sites formation and the resolution of AP sites, executed by multiple enzymes. DNA glycosylation enzymes cut off the *N*-glycosyl bonds by hydrolase activity and remove specific damage bases to produce base-free sites (AP sites), which are further treated with endonuclease and ligase. Finally, the complete repair of the original DNA sequence is realized. There are five DNA ribozymes specifically recognizing oxidized bases in mammalian cells. They are divided into two families: the Nth family including OGG1 and NTH1, and Nei family including NEIL1, NEIL2 and NEIL3 [38]. These two families are named after homologous proteins in bacteria—endonuclease III (Nth) and endonuclease VIII (Nei) [39,40,41], and each have different mechanisms of de-base lyase reaction. OGG1 and NTH1 cut the DNA chain by β-lyase activity, producing 3’ dRP and 5’ P ends, whereas NEIL1/2 have βδ lyase activity, thus creating 3’ P and 5’ P in the chain gap [42]. 

The mechanism of NEIL3 catalysis needs further study. So far, 8-hydroxyguanine DNA glucosidase 1 (OGG1) is a DNA repair enzyme found mainly in mammals that identifies and cleaves 8-oxoG on genomic DNA. OGG1 corrects the occurrence of 8-oxoG through a series of complex and subtle repair pathways to maintain genome accuracy. In the process, OGG1 binds to DNA containing damaged bases and bends the DNA strands, from which 8-oxoG is exposed and encapsulated in an OGG1 highly conserved catalytic active pocket, which thus cuts off the *N*-glycoside bond [43,44,45]. Among them, the binding ability of the OGG1 active pocket to 8-oxoG is about 105 times higher than that of peripheral pocket to guanine, which means the damaged base first enters the secondary center after OGG1 has been extruded into a double helix and then inserts itself into the active pocket (Figure 2a) [37,46,47,48]. This explains why OGG1 can accurately distinguish between highly similar 8-oxoG and G. The mechanism of OGG1 repair of 8-oxoG is summarized as follows: OGG1 removes the *N*-glycoside bond of the damaged base through the above recognition process, acting on a sugar-phosphate backbone to form an apurinic/apyrimidinic (AP)-site(s) (Figure 2b). Then, using its own weak AP lyase, OGG1 shows β-elimination thereby forming 5’ P (phosphate) and 3’ d RP (unsaturated hydroxyl aldehyde) ends. Because the 3’ end residue does not make a normal connection, OGG1 recruits apurinic/apyrimidinic endonuclease 1 (APE1) to clean up the 3’ end and get 3’-OH ends. At this point, DNA polymerase β inserts the correct nucleotide through the polymerization and DNA ligase III, and DNA ligase I connects the DNA chains, completing the base excision repair initiated by the OGG1 [42].

## 3. The Roles of Base Excision Repair Enzyme OGG1 in Gene Expression

Guanine is readily oxidized, thereby disrupting the integrity of the genome. From an evolutionary perspective, vertebrate genomes contain more guanine than other organisms. OxiDIP-Seq of human MCF10A cells shows that 8-oxoG is significantly enriched in the intergenic region and the intron region of gene body, and most of the 8-oxoG are enriched in protein coding DNA [49]. In total, 72% of human promoter regions are spread with higher GC base pairs [50], including gene-coding pro-inflammatory factors, proto-oncogenes and growth factors. Some conserved sequences recognized by transcription factors are also enriched with guanine. For example, transcription factor specificity protein 1 (SP-1) recognizes GC-rich sequences 5’-GGGGCGGGG -3’, and κB sequences identified by NF-κB have a 5’-GGGRNYYYCC-3’ pattern, of which 5’ ends are distributed with continuous guanine [51]. In recent years, there is growing evidence that 8-oxoG and its specific repair protein OGG1 have epigenetic regulatory effects on gene transcription. Some researchers found that Ogg1-/-mice shows significant 8-oxoG accumulation; however, it does not lead to cancer or affect life span and embryo development. Surprisingly, the inflammatory response in Ogg1-/-mice was also slighter [52], indicating that OGG1 function is closely related to promoting inflammatory cell recruitment and promoting cytokine expression.

Experimental evidence was widely reported in recent years. Researchers found an abnormal increase in 8-oxoG accumulation and a high spontaneous rate of lung cancer in the genome of Ogg1-/-mice [52]. The reduced immune response of Ogg1-/-mice models to endotoxic shock, diabetes and hypersensitivity manifested as decreased neutrophil infiltration and lower expressions of Th1 and Th2 cytokines [53]. Moreover, inflammation of the Ogg1-/-mice stomach was slighter after H. pylori infection when comparing to wildtype. Similarly, in an asthma model, the cell infiltration in the airway, perivascular and alveoli and the secretion of Th1, Th2 and Th17 cytokines of Ogg1-/-mice were slighter [54]. 

The small molecule inhibitor of OGG1,TH5487 is an effective selective active site binding agent, which inhibits the activity of OGG1 DNA glycosylase, so OGG1 cannot insert itself into the DNA chain and join the 8-oxoG, and also reduces the DNA occupation of NF-κB and the level of TNFα-induced neutrophil recruitment and various cytokines in human bronchial epithelial cells and in mice models [55]. In 2019, Pan et al. [56] showed that in lung inflammation challenged by ozone, levels of 8-oxoG in genomes gradually ascend. HE staining showed that with the prolongation of ozone stress, more inflammatory cells infiltrated airways around the alveoli, and other manifestations such as airway stenosis, partial alveolar rupture and enlarged alveolar cavity arose. After five days of ozone stress, 8-oxoG accumulated in the airway epithelium and showed an increased oxidation level of protein. The OGG1 inhibitor TH5487 treatment significantly improved above phenomenon and manifested as reduced infiltration of inflammatory cells in the airways, surrounding alveoli and the complete alveolar structure. There are also other studies reporting small molecular inhibitors designed for OGG1, which provide a bright outlook for clinical anti-inflammatory treatment [57].

The aforementioned studies suggest that OGG1 has immunomodulatory functions, which are closely related to promoting inflammatory cell recruitment and cytokine expression. Studying the role of 8-oxoG and its repair enzyme OGG1 in pulmonary inflammation plays an important role in elucidating the mechanism of pulmonary inflammation and developing new therapeutic targets for chronic pulmonary inflammation from the perspective of oxidative stress. 

Further studies show OGG1 plays a role in regulating gene transcription, called DNA repair coupled gene transcription [58]. Genome-wide ChIP-Seq results indicate that OGG1 significantly recruited in the gene regulatory regions, such as promoters, introns, exons, intergenic sequences and untranslated region in HEK293 cells stimulated by TNFα [59], suggesting that OGG1 could conduct its regulatory function in wide genome regions. So far, because of the different stimuli and target genes, there are several ways for 8-oxoG and its repair enzymes OGG1 to regulate gene transcription [60,61,62,63], and the specific mechanism of OGG1 in regulation of different genes still calls for further study.

### 3.1. OGG1 Regulates Gene Expression via Nonenzymatic Pathway

OGG1 can prompt inflammatory gene transcription without depending on its enzymatic activity. It could be an epigenetic mechanism to regulate innate immune responses by non-excisional binding to 8-oxoG in promoter sequences [64]. Stimulated by TNF-α, cells produce large amounts of ROS, leading to an accumulation of 8-oxoG. Meanwhile, OGG1 cysteine oxidizes, which does not affect the ability to identify 8-oxoG but inhibits its cleavage activity. Binding to 8-oxoG in the promoters of CXCL1 and TNF-α and changing the DNA structure thus recruits transcription factors such as NF-κB to bind to their cis-acting elements rapidly, which can promote the rapid expression of inflammatory genes. With the addition of NAC (ROS scavengers) or the interference of OGG1, both the ability of NF-κB to bind to promoter regions and the expression of inflammatory genes decreased significantly [65,66]. Among them, Cxcl2, a gene related to aging and chronic inflammatory states, is at the highest level in a series of chemokines and cytokines. Therefore, OGG1 could be a potential target for aging-related diseases and chronic inflammation such as asthma, COPD and so on. NF-κB family mainly includes five subunits: RelA/p65, RelB, c-Rel, p50 (its precursor is p105/NF-κB1) and p52 (it precursor is p100/NF-κB2). NF-κB must act in the form of homologous or heterodimers. RelA/p65, RelB and c-Rel contain RHR domains with transcriptional activation; therefore, there must be at least one subunit in the dimer where it can have transcriptional activity. The heterodimer p50/p65 is a classical transcription factor in Rel family that regulates many cellular functions, such as immune response, cell growth and development. 

Researchers found that there is a bidirectional interaction between OGG1 and NF-κB. OGG1 influences the combination of NF-κB to DNA through the combination of 8-oxoG, and the interaction between OGG1 and p50R sequence is more abundant. OGG1 increases the binding of NF-κB to DNA through the combination with 8-oxoG outside the κB motif, whereas the combination with 8-oxoG within κB motif reduced the binding. OGG1 not only promoted the recognition of NF-κB to conserved sequences, but also shortened the time of the recognition, whereas NF-κB could also increase OGG1-DNA binding. Through these interactions, OGG1 ultimately promotes cxcl2 expression of TNF-α stimulated cells by activating promoter activity. This plays an important role in the rapid expression of specific genes under stress conditions [65,66].

The regulatory model of gene transcription by OGG1 via its non-enzymatic pathways can be described as below: When an organism suffers from endogenous respiratory metabolism, products or environmental pollutants, microbial infections and a massive number of ROS product, on the one hand transcription factors are activated, and on the other hand, a lot of 8-oxoG is generated. Then, 8-oxoG is not repaired immediately; instead oxidized OGG1 bound to DNA promoter regions is recruited. OGG1 invades the DNA helix through amino acid residues, extracts the damaged base and inserts it into the OGG1’s catalytic center pocket, whereas the cytosine paired with 8-oxoG remains unmoved in the DNA helix. These events eventually cause violent bending of DNA double strands at the angle of 70° at the action site and adjacent sequences, resulting in structural changes [46,67], which enable activated transcription factors to rapidly recruit to the corresponding promoter region (perhaps this double helix overcomes the energy barrier required for NF-κB to bend DNA), and then recruit other proteins to form transcription initiation complexes and initiate gene expression. Meanwhile, ROS can be cleared through an antioxidant system to keep redox at equilibrium. OGG1 initiates a base excision repair pathway after reduction and ensures the stability of the genome [68].

In addition, Pan et al. showed that OGG1 can regulate the expression of tissue inhibitors of metalloproteinase 1 (TIMP-1) in lung tissue through a nonenzymatic pathway [56]. TIMP-1 plays an important role in pulmonary fibrosis, which can regulate cell proliferation, apoptosis, differentiation and angiogenesis. The regulatory sequence of TIMP-1 is located in the intron 1 [69], which contains a large number of 5’ GGG (even 5’ GGGGGGG), making it easy to produces 8-oxoG under oxidative stress. As a result, it is a potential binding target for OGG1. The authors proposed the following models: In the intron region, when transcriptions enter the extension phase, there are DNA: RNA hybrids with about 11 nucleotides inside the transcription machine. The nascent RNA are transcribed firstly in the chromatin, then mature mRNA are released after a selective shearing. OGG1 can inhibit the release of the nascent RNA by binding to the DNA: RNA hybrids, thus reducing the mRNA of TIMP-1. The specific role of OGG1 in regulating TIMP1 in pulmonary inflammation and fibrosis needs further experimental proof. In addition, whether this mechanism is applicable other cytokines also needs further study. 

To sum up, similar to DNA methylation modification (5 mC), which was regarded as a classical epigenetic marker, DNA oxidative modification (8-oxoG) also has epigenetic characteristics and can change the process of gene expression. OGG1 has a regulatory function in the early stage of transcriptions. Epithelial cells are first subjected to oxidative stimulation, then OGG1 recognizes oxidative-damaged bases produced within gene regulatory regions, changing the pattern of gene transcription. Expression changes of cytokines or chemokines in epithelial cells form the basis of pulmonary inflammation initiation, indicating a bright prospect for the design of small molecular inhibitors of OGG1 intended for inflammation control.

### 3.2. OGG1 Regulates Gene Expression via Enzymatic Activity-Dependent Pathway

#### 3.2.1. OGG1-BER Mediated G-Quadruplex Regulation of Gene Expression

OGG1 can modify the DNA structure through the AP sites and productions of the BER, to promote gene transcription. Hypoxia is a very common pathological process in clinical diseases, involving the expression of genes such as heat shock factors, glycolytic enzymes, extracellular matrix factors, cytoskeletal factors, apoptotic factors, cell cycle regulators and angiogenic factors [70]. Hypoxia can induce lung inflammation, in which HIF1α plays an important role. Under hypoxia conditions, ROS, produced by pulmonary endothelial cells, makes instant oxidative modifications to the hypoxia-response element (HRE) of vascular endothelial growth factor (VEGF), further adjusting the expression of VEGF [71]. Further research finds that under hypoxia conditions, ROS produced by mitochondria leads to the accumulation of HIF-1, the transcriptional regulators of hypoxia gene expression. But when hypoxia-induced base modification is suppressed, the binding of HIF-1 to HRE of the promoter of VEGF is weakened, and the mRNA of VEGF also decreased. Pastukh V proved that this process was related with 8-oxoG by chromatin immunoprecipitation of pulmonary artery endothelial cells [72]. Inhibition of OGG1 by siRNA decreased the hif-1α binding to Ref-1/Ape1 in each HRE region, thus inhibiting the expression of VEGF mRNA. Among them, HIF-1 interact with Ref-1/Ape1, but not OGG1.

G-quadruplex is a special nucleic acid structure, whose constituent unit is G-quartet. G-quartet is the square planar structure formed by the binding of multiple guanines by Hoogsteen hydrogen bonds. With cations (e.g., Na, K) coordination, two or more G-quartets form G-quadruplex by π-π stacking. The human genome bioinformatics analysis found that more than 300,000 genes may form G-quadruplex [73]. The location of G-quadruplex is nonrandomized, and it co-localizes with functional regions of the genome, especially in telomeres and promoters, and is highly conserved among species. In addition, G-quadruplex is positioned at the 5’ UTR, 3’ UTR, intron-exon boundary, especially in the first intron, suggesting its broader function [74].

Current research shows that G-quadruplexs play roles in transcription, replication and other key biological processes, such as hereditary diseases and cancer. Reportedly, genes under regulation of the promoter PQS include VEGF, PDGF-A, KRAS, HRAS, SRC, etc. In addition, PQS are rich in promoters of human DNA repair genes, suggesting the possibility of its regulation on those genes. OxiDIP and Quadron analysis in MCF-10A cells found that 37% of the 8-oxoG peaks contained potential G4 structures [49], most of which have high folding potential, suggesting an unrevealed link between 8-oxoG and G-quadruplexs.

Specific G-quadruplexs formed in the VEGF promoters and VEGF gene transcription can be controlled by ligand-mediated G-quadruplex stabilization [75]. The production of VEGF may be inhibited in tumor cells via targeting the G-quadruplexs formed in guanine-rich-regions in VEGF gene promoters. Fleming AM further elucidated these findings, suggesting that ROS-mediated production of 8-oxoG within promoters is a signal transduction agent for gene activation [76]. Large quantities of 8-oxoG are generated under ROS, and the formation of AP sites during OGG1-initiated-BER leads to DNA structural instability. Then, AP sites are preferentially squeezed into a ring and form the G-quadruplex (G4), which is a four-strand spiral structure. At this point, APE1 binds to AP sites to recruit transcription factors to promote gene transcription instead of cutting. When G4 forms on the template chain, it inhibits gene expression [77,78,79]. Whether OGG1-BER regulation to gene expression is applicable to other hypoxia-related genes still calls for ChIP-seq and bioinformatics analysis.

The 8-oxoG in the human KRAS gene is more abundant in the G4 region than the non-G4 region. 

Normally, G4 sequences exist in the promoter regions of genes related to cancers such as KRAS and HRAS, thus inhibiting transcription. The high metabolic rate of cancer cells leads to increased ROS levels, which facilitate the oxidation of guanine. 

Attacked by ROS, cells generate 8-oxoG, which is more likely to occur inside the G-quadruplex, and the zinc finger transcription factors MAZ, SP1 and heterogeneous ribonucleoprotein A1 will bind to the G4 quadruplex to unfold it to form a DNA double helix, recruiting OGG1 to initiate the BER pathway to ensure the accuracy and integrity of the genome, which allows the recruitment of transcription factors and RNA polymerase II to promote gene transcription [80,81].

#### 3.2.2. OGG1-BER Recruits Topoisomerase to Promote Gene Expression

The hormone estradiol 17β (E2) and homologous estrogen receptors (ERα and ERβ) bind to DNA elements with high affinity, promoting expression of genes related to cell cycle progression and apoptosis [82] and controlling the growth and survival of hormone-sensitive cells. In this process, after lysine-specific demethylase1 (LSD1) binds to ERα, it mediates the demethylation of H3K9me2 within the enhancer and promoter regions, which leads to the production of H_2_O_2_. The Fe(II) or Cu(I)-mediated Fenton reaction could activate H_2_O_2_, thereby oxidizing G to OG, and causing the accumulation of 8-oxoG [83]. This accumulation is related to the presence of ERa and the activation of LSD1. Using the LSD1 inhibitor monoamine oxidase or reducing the expression of LSD1 can reduce the production of 8-oxoG [84,85,86]. The 8-oxoG mediated by LSD1 is first targeted for removal by the base excision repair system initiated by OGG1. This repair process leads to single-strand breaks and transient gap generation, which become the substrate of topoisomerase II (TOPO). The recruitment of topoisomerase II can lead to changes in DNA structure, promote chromatin accessibility or DNA bending to help RNA polymerase II loading onto its target genes, thereby contributing to transcriptional activation (such as Bcl-2).

Cell death processes can control lung inflammation. In response to DNA damage or endoplasmic reticulum stress, when the balance of pro-apoptotic and anti-apoptotic mediators of Bcl-2 family proteins reach the level that is conducive to cell death, they will mediate cell death through intrinsic or mitochondrial pathways. Understanding the mechanisms regulating pulmonary cell death will help to identify new therapeutic targets to reduce or cure lung inflammation [87].

### 3.3. OGG1 Regulates Gene Expression through Chromatin Modification

The modification of DNA and chromatin plays an important role in gene epigenetic regulation. Studies have shown that OGG1 can affect gene expression by recruiting chromatin restructuring complexes. CHD4 is a key component of the complex of nucleosome remodeling and histone deacetylation (NuRD). It is essential for DNA damage repair (DDR) and is related to carcinogenesis (including abnormal stem cell renewal and dullness, differentiation and changes in cell cycle regulation) [88]. The role of CHD4 in DNA double-strand breaks is to recruit inhibitory chromatin to the open chromatin region of the active gene promoter and protect the transcription region during the repair process [89,90]. Under normal circumstances, the methylation level of CpG islands in the regulatory region of tumor suppressor gene (TSG) expressed in human colon cancer cells is very low, and the loose chromatin structure is conducive to gene transcription. 

When cells are under oxidative stress, OGG1 can bind to 8-oxoG and further recruit chromodomain-helicase-DNA-binding protein 4 (CHD4). On the one hand, CHD4 will recruit DNA methyltransferases (DNMT1, DNMT3B) to promote the methylation of cytosine and make this region highly methylated; On the other hand, CHD4 will recruit histone methyltransferases (EZH2, G9a) to methylate histone H3, forming the trimethylation modification of H3K27 (H3K27me3) and the dimethylation modification of H3K9 (H3K9me2), thereby inhibiting gene transcription. 

After suppressing the expression of OGG1, researchers found that the content of 8-oxoG in the genome increased, but CHD4 could not be recruited to the promoter region of tumor suppressor genes, indicating that OGG1’s recognition and combination of 8-oxoG resulted in lower tumor suppressor genes expression. When cells are under long-term oxidative stress, the expression of tumor suppressor genes will be inhibited, which may be related to the occurrence of cancer [91].

A recent study confirmed the role of OGG1 in the symmetric dimethylation of histone H4-arginine-3. OGG1 binds to its DNA substrate to recruit arginine *N*-methyltransferase 5 (PRMT5), which can catalyze the symmetric demethylation of arginine-3 and cause gene silencing [92].

### 3.4. OGG1 Regulates Gene Expression by Forming a Complex with Free 8-oxoG

In addition to the aforesaid mechanisms of gene expression regulation, there are studies that point out that the complex formed by OGG1 and free 8-oxoG can affect gene expression.

OGG1 has high affinity with free 8-oxoG base (the binding constant Kd is 0.56 n M), and the complex formed by binding to the nonsubstrate site of 8-oxoG has a guanine-nucleotide exchange factor (GEF) activity, which does not exist in cells lacking OGG1. The OGG1-8-oxoG complex can activate canonical Ras family GTPases (canonical Ras family GTPases), such as K-RAS (Kirsten rat sarcoma viral oncogene homolog), N-RAS (Neuroblastoma RAS viral oncogene homolog) and H-RAS (Harvey-RAS), and replace its GDP with GTP [93], inducing the phosphorylation of its downstream targets Raf1, MEK1,2 and ERK1,2, further participating in cell signal transduction, and play a role in cell inflammation and pathophysiological processes.

The members of the small GTP binding protein superfamily are structurally divided into at least five subfamilies: Ras, Rho, Rab, Sar1/Arf and Ran families. Among them, the Ras subfamily members (Ras protein) mainly regulate gene expression. The Rho/Rac/Cdc42 subfamily members (Rho/Rac/Cdc42 protein) of the Rho family regulate cytoskeletal reorganization and gene expression [94]. Rac1 is mainly expressed in lung tissues, especially non-phagocytic cells, such as lung epithelial cells and fibroblasts. The OGG1-8-oxoG complex physically interacts with GDP-bound Rac1, and GDP is quickly converted to GTP through GEF activity, thereby increasing the GTP-bound Rac1, and Rac1-GTP can give a transient rise to ROS levels and participate in a series of cell signaling transduction. The specific biological significance of OGG1-BER-related ROS production is not known yet. Experiments have found that locally increased ROS can oxidatively modify the cysteine residues at the active site of OGG1, thereby reducing the cleavage activity of OGG1 [95]. Further observation found that the non-productive combination of OGG1, which temporarily lost its cleavage activity, and 8-oxoG can increase the expression of NF-κB-dependent inflammatory factors (such as CXCL2) [64].

The OGG1-8-oxoG complex can also increase the level of Rho-GTP in cells (such as fibroblasts and epithelial cells), thereby mediating the polymerization of α-smooth muscle actin (α-SMA) into stress fibers and increasing the level of α-SMA in insoluble cells/tissue and participating in lung remodeling and fibrosis (including idiopathic pulmonary fibrosis) [96]. Researchers speculate that the reduced OGG1 activity observed in fibrotic tissues and malignant cells may be a cellular defense against the extensive cytoskeletal changes required for cancer cell proliferation and migration.

## 4. Roles of OGG1 in Pulmonary Inflammation and Disease

### 4.1. The Roles of OGG1 in Lung Cancer

Lung cancer ranks as one of the cancers with the highest morbidity and mortality in recent years, which is also widely reported to have close association with oxidative stress. Interestingly, there have been studies over the role of OGG1 in lung cancer separated into different directions at the very beginning. 

Case-control studies carried out in populations of different diseases, regions or life states, together with meta-analysis based on them, tried to figure out the correlation between OGG1 and lung cancers, and the conclusions were inconsistent. Some researchers summarized that OGG1 has no relation to a higher risk of lung cancer [97,98], or if the association appeared in certain populations, such as in non-smokers [99]. Opposite opinions suggest that the Cys/Cys genotype of the OGG1 Ser326Cys polymorphism was believed to have an association with a lung cancer risk [100], and the combined OGG1-Cys/Cys and Ser/Cys genotypes show a 1.93-fold increased risk of lung cancer, which was particularly elevated among women who suffered from relatively high cumulative exposure to smoky coal [101]. Besides, genome-wide association studies (GWASs) found that the genotypes for two DNA repair genes, TP53 and OGG1, showed significant associations with lung squamous cell carcinoma (SQC) risk [102].

Studies over the impact of OGG1 on lung cancer also did not reach a consensus. Researchers reported that TET1 regulated the balance of hydroxymethylation and methylation in the promoter region of BER related genes, which participates in the development of lung cancer induced by environmental chemicals [103], and in the tumor tissues of smoking patients with non-small cell lung carcinoma. The activity of hOGG1 was significantly higher compared with control samples, whereas the activity of endonuclease III homologue (hNTH1) was lower. Such alterations could affect tumor growth by increasing the number of AP sites [104]. These aforesaid results imply that OGG1 may contribute to tumorigenesis.

On the contrary, it was observed that OGG1 mutant mice manifested an increased susceptibility to the multiorgan carcinogenesis induced by *N*-diethylnitrosamine (DEN), *N*-methyl-*N*-nitrosourea (MNU), *N*-butyl-*N*-(4-hydroxybutyl) nitrosamine (BBN), N-bis (2-hydroxypropyl) nitrosamine (DHPN) and 1,2-dimethylhydrazine dihydrochloride (DMH) (DMBDD) [105], and OGG1 depletion suppressed A3 T-cell lymphoblastic acute leukemia growth, both in vitro and in vivo, suggesting that OGG1 could play a role as a potential anti-cancer target [106]. As corroborative evidence, other research found that the CDK4/6 inhibitor sensitized G1-arrested cells to anticancer drugs by downregulating the expression of PARP1, on which OGG1 depends to conduct its DNA damage repair function [107]. Additionally, the anticancer effects of MTH1 inhibition also require the existence of OGG1, transforming the mismatch caused by 8-oxoG to the DNA strand break to injure oncocyte [108].

### 4.2. The Roles of OGG1 in Innate Lung Immunity

Airway epithelium is the main surface in contact with inhaled particles, pathogens and allergens, and is lined with a semi-impermeable barrier of highly adapted epithelial cells [109]. Epithelial cells play a central role in triggering the protective host response. After microbial invasion, airway epithelial cells first activate the reproductive system coding Pattern recognition receptor, recognize the pathogen-associated molecular pattern (PAMPs) or damage-associated pattern (DamPs), and then trigger cells to produce innate immunity (IIR) to prevent/reduce the spread of foreign pathogens, which is then key to triggering adaptive immunity [110].

Ros produced in the PRR signaling pathway plays an important role in signal transduction by controlling phosphorylation. Little is known about the synergistic effects in IIR of cellular reactive oxygen species (ROS). It was proposed that OGG1 and Ataxia telangiectasia-mutated (ATM) are endogenous nuclear ROS sensors that coordinate the Pattern recognition receptor signaling pathway and regulate the innate immune response [95]. ATM responded to DSBs/ROS, forming a scaffold with ribosomal S6 kinase, which induced RelA phosphorylation and resulted in the transcription coupling of type I and III IFN, CC and CXC chemokines. The cytosolic OGG1-8-oxoG complex acts as a guanine nucleotide exchange factor, inducing the formation of MAP-, PI3-and MS-kinases, and activating the classical NF-kB pathway by phosphorylation and nuclear translocation at Ser276, and playing a role in innate immune response (IIR).

### 4.3. The Roles of OGG1 in Airway Remodeling and Asthma

Genes under regulation of the OGG1-8-oxoG complex are associated with some important biological processes and airway remodeling. Researchers performed an imitation of the OGG1-BER process by attacking mice lungs using 8-oxoG. In total, 1592 transcripts were identified by RNA-seq analysis [103]. The up-regulated mRNA was associated with biological processes, including homeostasis, immune system, macrophage activation, surface tension regulation and response to stimuli. These processes are mediated by chemokine, cytokine, gonadotropin-releasing hormone receptor, integrin and interleukin signaling pathways. In addition, their analysis of another study identified 3252 differentially expressed transcripts, of which 2435 were up-regulated and 817 were down-regulated [102]. In the up regulated transcripts, 2080 mRNA were identified, encoding proteins involved in the regulation of actin family cytoskeleton, extracellular matrix, cell adhesion, cadherin and cell junctions that influence biological processes such as tissue development, cell-to-cell adhesion, cell communication and the immune system. OGG1-BER involved in the overexpression of cadherin, integrin, Rho GTP enzyme, TGF, WNT and cytokine/chemokine signaling pathway indicates that OGG1-BER can continuously repair DNA oxidative damage and conduct downstream signal transduction through small GTP enzyme and induce airway remodeling associated with gene expression, leading to changes in lung function and structure.

Asthma is characterized by airway inflammation and hyperresponsiveness, and the tilt towards TH2 cytokines is considered to be a key risk factor for asthma. There is growing evidence that oxidative stress is also a key factor in the development of asthma. Oxidative stress levels in the lungs of asthmatics are increased and ROS produced in the cells regulates the gene expression of asthma-associated TH2 cytokine IL-4 [111]. In addition, oxidative DNA damage in the peripheral blood lymphocyte was significantly higher in asthmatic patients than in healthy subjects [112]. Whole-genome expression analysis also suggests that the OGG1-BER-induced gene expression may play a role in asthma and EIA Pathophysiology [113,114]. OGG1 was found to up-regulate the expression of Cytokines, especially IL-4, through the STAT6/NF-κB pathway in OVA-sensitized asthmatic mice [54].

### 4.4. The Roles of OGG1 in Allergic Airway Inflammation

Ragweed pollen extract (PWPE), which contains NADPH Oxidase (Nox; RWPENOX), can induce the production of a reactive oxygen species (ROS) and lead to airway hypersensitivity. Bacsi et al. found that the level of 8-oxoG increased in the airway epithelial cells of mice challenged with RWPE, and DNA SSB formed during 8-oxoG repairment enhanced antigen-driven allergic immune response [115]. It was found that attacking OGG1 proficient mice with ragweed pollen induced strong recruitment of airway eosinophil granulocyte, whereas attacking OGG1 deficient mice showed reduced recruitment [116]. After infusing free 8-oxoG into mouse lungs to simulate the OGG1-BER pathway and evaluate the unbiased RNA sequencing and molecular histologic changes, researchers found that PWPE attack could induce an oxidative burst, cause DNA damage and activate the OGG1 signal, resulting in the differential expression of 84 microRNAs (miRNAs). OGG1 can increase levels of TH2 cytokines (such as IL-13, IL-4 and IL-5) by downregulating miRNA let-7b-3p, which leads to Eosinophil granulocyte recruitment and exacerbates allergic airway inflammation. Affymetrix microarray genechips show that in airway epithelial cells, the OGG1-Ras regulatory network is regulated by TF, including NF-κB, tumor protein-53 (TP53), Kruppel-like factor 4 (KLF4) and so on [117]. It is suggested that the recruitment of OGG1-BER-small GTPases, activated TF and/or OGG1-mediated recruitment of TF in promoter regions may be involved in miRNA transcription regulation. The aforesaid research suggest that it is worth exploiting a new therapy to alleviate airway inflammation achieved by pharmacological modulation of OGG1 signal transduction or local administration of specific miRNAs.

### 4.5. The Roles of OGG1 in Hyperoxia-Induced Lung Injury

In clinical work, oxygen was given as a supportive treatment for patients suffering from acute respiratory distress syndrome (ARDS). However, hyperoxia could also induce lung injury. Considering its function as a key factor in the response to oxidative DNA damage, it is natural to consider of the role of OGG1. The existence of OGG1 could help elevate the resistance to hyperoxic cytotoxicity, overexpression of hOGG1, alleviate DNA damage of A549 and AECII from hyperoxia and H_2_O_2_ exposure, and could be related to the MAPK pathway activation [118,119]. It has been reported that inflammatory cytokines (TNF-α,IL-6,IFN-γ) increased in the OGG1 deficiency in mice after hyperoxia exposure [120], as OGG1 interacted with the promoter of Atg7, thus regulating the autophagy pathway to influence inflammatory cytokine release in the hyperoxia-induced lung injury.

Oxidative stress has been testified to be a non-negligible risk factor for neonatal disease, especially affecting pulmonary development. Supplementary oxygen for respiratory disfunction in infants is an important source of oxidative stress, causing lung injury that manifests as delayed alveolar growth, extracellular matrix deposition and pulmonary fibrosis, which could evolve into permanent lung injury—bronchopulmonary dysplasia (BPD). Human lung morphogenesis is divided into five stages: embryonic, pseudoglandular, canalicular, saccular and alveolar. Premature infants are under a higher risk of being attacked by supplementary oxygen, since their lungs tend to be at the saccular-alveolar stage, during which lungs are more sensitive to oxidative stress [121]. Elevated 8-OHdG levels along with lung injury were observed in a hyperoxia model, and its alleviation after suppressing oxidative stress implied the potential role of OGG1 in pulmonary development [122], which was subsequently documented by another research team, who suggested that OGG1 expression was upregulated in a hyperoxia-induced BPD model [123]. Interestingly, the expression of OGG1 showed a time-related fluctuation, which reached a peak at a certain timepoint but descended to normal when hyperoxia exposure was prolonged.

## 5. Conclusions

The breakdown of redox homeostasis brings multiple organs’ diseases to the human body, among which, pulmonary diseases were closely related to DNA oxidative damage. The ROS-overproduction-induced accumulation of base mismatch has been proven to be one of the pathogenic factors of inflammation, fibrosis and tumorigenesis. Base excision repair initiated by OGG1 takes the majority of the responsibility for the correction of mismatched base pairs, since 8-oxoG is the most common type generated from oxidative attack. As the investigation over OGG1 gets deeper, research over the other perspective of OGG1 function were reported. As the big picture became complete, scientists found that during the whole process of BER, from recognition to excision, even its combination with free 8-oxoG, OGG1 impacts gene expression in multiple aspects.

With the functions originated from its DNA binding characteristic come the conflicts that need to be solved. Take inflammation for example; in an hyperoxia-induced lung injury model, despite the fact that we expect OGG1 to conduct a pro-inflammation function, researchers found that overexpression of OGG1 and Fpg protected alveolar epithelial cells from toxics in oxygen-derived species [119]. Meanwhile, OGG1 deficiency mice exhibited a significant increase of pro-inflammatory cytokines (TNF-α, IL-6, IFN-γ) after challenged by hyperoxia [120], indicating a positive role of OGG1 in hyperoxia-induced lung injury. 

Moreover, when another perspective was taken into consideration, alleviation of inflammation is also a promising target for BPD treatment. Inhibition of the NF-κB pathway by either Grx1 ablation [124] or Quercetin [125] treatment shows protective effects on pulmonary angiogenesis and alveolar growth in variant degree. As Pan Lang documented, after oxidative stimulation, the precedingly binding of OGG1 to the 8-oxoG upstream from NF-κB motif in the promoter regions of pro-inflammation gene increased NF-κB DNA occupancy and gene expression, whereas depletion of OGG1 shows the reverse effect on both NF-κB binding and gene expression. Thus, it creates a puzzle about what the overall effect of OGG1 to inflammation is when confronting oxidative stress.

In summary, studies performed so far have given enough evidence that OGG1 plays important role in pulmonary diseases and show the necessity of further investigation on its relative mechanism. That is to say, problems remaining unsolved are also part of our expectations.

## Figures and Tables

**Figure 1 antioxidants-11-00933-f001:**
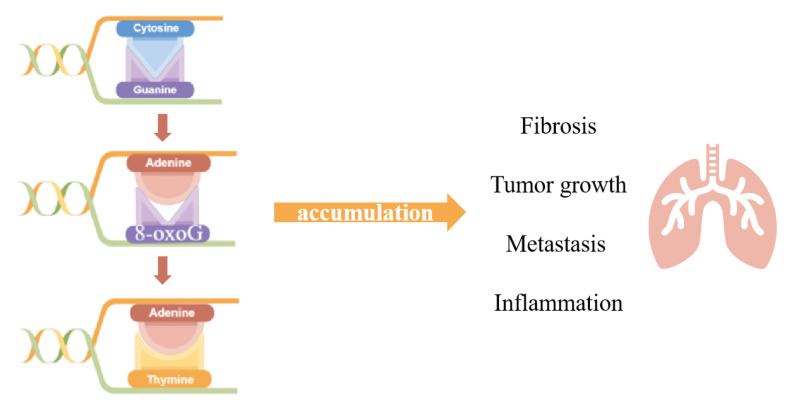
After oxidative stress attack, 8-oxoG is the most common form of bases which could mismatch with Adenine. The Adenine mismatched with 8-oxoG would match with Thymine after the normal DNA replication cycle, generating the transformation from GC-TA. Accumulation of such a transformation could lead to pathological processes, such as fibrosis, tumor growth, metastasis and inflammation. (drawn by Figdraw).

**Figure 2 antioxidants-11-00933-f002:**
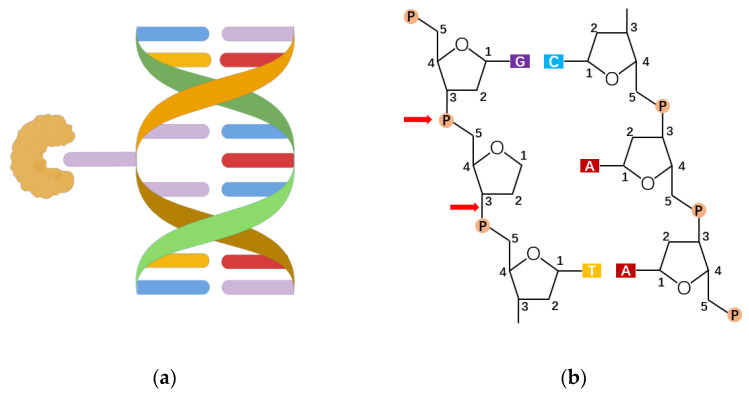
The recognition and excision of 8-oxoG. (**a**) The lesion-specific binding pocket of OGG1 allows the recognition of the extrahelical flipped 8-oxoG, and excises it from DNA strand (drawn by Figdraw). (**b**) OGG1 acts on a sugar-phosphate backbone to form AP-sites, generating 5’ P ends and 3’ d RP ends.

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
