# Peer review of "OGG1 in Lung—More than Base Excision Repair"

_antioxidants, 2022, doi:10.3390/antiox11050933_

Round 1

Reviewer 1 Report

In the present review, the authors described the role of OGG1 in regulating gene expression associated with oxidative stress. The authors also showed the involvement of OGG1 in several pulmonary diseases. The review is well-described with intriguing studies. I have some minor comments to be addressed before publication.

  1. Some of the sentences sound a bit strange to me. English proofreading should be needed.
  2. In Figure 1, the letters in the last base with adenine were not clear enough without any explanation.
  3. Some of the references were from the quite low impact factor journals without any other replication studies. Please change the references into the confirmed ones.
  4. Please correct the typos.

Reviewer 2 Report

The biochemical activity of 8-oxoguanine DNA glycosylase (OGG1) has been extensively studied, revealing that this enzyme recognizes 8-oxoguanine base pairs, catalyzes expultion of the aberrant base and cleaves the DNA backbone. 8-oxoguaine is considered to be one of the major endogenous mutagens contributing to spontaneous cell transformation and plays a  pivotal role in human cancers. OGG1 is called repair enzyme and is very important for maintaing redox homeostasis.  In the light of that reviewer believes that the present review entitled "OGG1 in lung-more than base excision repair" is an important contribution. It will help  to  understand better the mechanisms through which OGG1 regulates gene expression and is involved in pulmonary diseases.                                                               Major criticism:  The reviewer  would add the full name of OGG1 into the  abstract section.  The chapter: 4.4 is terminated by a sentence:...... may have clinical effects. It needs more specific explanation: what kind of clinical effects authors have in mind? There are a lot of spelling mistakes or misspeling: page 2, 3, 4, 5, 11, 12 (capitol letter in the middle of the sentence, ie.  ....Organisms, ....Facing- p.3). References: page 5, row 7 from above - there is : "Pan L" should be "Pan et al." and the number of reference; page 6, the same; page 11, row 18 from below - there is "Attila Bacsi", should be "Bacsi et al."  Other misspeling: it should be  G-quadruplex, page 12 , row 19 from above- sentence "Higher risks of ......" does not make any sence-  needs to be corrected. Page 13, the frase  : "........ previous studies ...."  shoud be replaced by ..... "studies performed so far...".  In summary, this is an interesting review which the reviewer recommends for publication after minor revision.
